# Granulocyte Apheresis: Can It Be Associated with Anti PD-1 Therapy for Melanoma?

**DOI:** 10.3390/medicina58101398

**Published:** 2022-10-06

**Authors:** Alvise Sernicola, Anna Colpo, Anca Irina Leahu, Mauro Alaibac

**Affiliations:** 1Dermatology Unit, Department of Medicine (DIMED), University of Padua, 35122 Padua, Italy; 2Apheresis Unit, Department of Transfusion Medicine, University Hospital of Padua, 35128 Padua, Italy

**Keywords:** melanoma, granulocyte and monocyte apheresis, immunotherapy, immune checkpoint inhibitor, neutrophils, PD-1, tumor microenvironment

## Abstract

In the field of advanced melanoma, there is an urgent need to investigate novel approaches targeting specific components of the cancer–immunity cycle beyond immune checkpoint inhibitors. The authors reviewed the basic understanding of the role of neutrophils in cancer biology, and the latest clinical evidence supporting the correlation between cancer-associated neutrophils and the prognosis and response to the immunotherapy of advanced melanoma. Finally, they propose that granulocyte and monocyte apheresis, an emerging non-pharmacological treatment in current dermatology, could become an investigative treatment targeting melanoma-associated neutrophils which could be potentially used in combination with the usual immune checkpoint inhibitors.

## 1. Introduction

The introduction of immunotherapy with immune checkpoint inhibitors has transformed the treatment scenario of advanced and metastatic melanoma with the achievement of sustained response and long-term survival rates that were unthinkable with classical oncologic therapies [1,2]. Melanoma is rightfully considered an immune-responsive malignancy: the development and progression of this cutaneous tumor is strongly driven by downregulators of the immune system, such as PD-1 and its ligand, that constitute cancer mechanisms for immune evasion. Inhibitors of the PD-1 immune checkpoint proved able to overcome immune escape and reestablish the antitumor activity of melanoma-infiltrating T lymphocytes [3].

However, patients with advanced or metastatic melanoma (as well as those with other solid neoplasms) may not respond to this systemic treatment in up to 50% of cases and can benefit from limited therapeutic alternatives [4]. Effective approaches may be further narrowed in cases of elevated cancer burden and of multiple secondary localizations of malignancy. These limitations created an urgent need to investigate novel approaches to target additional specific components of the cancer–immunity cycle, which may heat up the tumor microenvironment (TME) promoting cancer cell recognition and boosting the effect of PD-1 inhibitors [5].

The authors have reviewed the current knowledge of the role of neutrophils in cancer biology, with a focus on melanoma, and the recent correlation of cancer-associated neutrophils with the prognosis and response to immunotherapy in advanced melanoma. Finally, they propose granulocyte and monocyte absorptive apheresis, an emerging non-pharmacological treatment in dermatology, as a future investigative treatment targeting melanoma-associated neutrophils that could be potentially used in combination with the usual immune checkpoint inhibitors.

## 2. Neutrophil Biology and Cancer

While it is established that neutrophils are the key players of the innate immune system, orchestrating infective and inflammatory responses, the study of their role in tumors is relatively recent [6]. Traditional neutrophil functions related to innate immunity are in phagocytosis, the release of granules containing cytotoxic factors and as neutrophil extracellular traps (NETs) [7]. Recent studies have highlighted that certain neutrophil subsets act as antigen-presenting cells, coregulators for T cells and antibody-dependent cytotoxic cells [8,9].

The developmental stages of neutrophils succeed one another during granulopoiesis, which takes place in the bone marrow: hematopoietic stem cells yield multipotent progenitors, followed by common myeloid progenitors, granulocyte and monocyte progenitors, pro-neutrophils or committed progenitors, neutrophil precursors, immature neutrophils, and mature cells [6]. Baseline neutrophil production has been estimated around one billion cells per kg daily and this value may increase by a factor of 10 following inflammatory stimuli [10]. At homeostasis, neutrophils display a differential tissue distribution, with high frequencies in the bone marrow and lower frequencies in organs such as the skin [11].

In the TME of melanoma, as in that of other tumors, the most represented immune cell population is that of neutrophils. Despite their abundancy, the molecular interactions in which they are involved have only been partially explored. It is known that cancer secretes the potent pro-inflammatory cytokines, TGF-beta, IL-6 and IL-8, that are able to recruit neutrophils, and drive a switch from an anti-tumor phenotype to a tumor-promoting and immunosuppressive profile [12,13].

The mobilization of neutrophils from the bone marrow is regulated by activating the CXCR2 chemokine receptor [10]. IL-8 is a ligand for CXCR2 that is secreted by several cancer types [14] and the activation of NF-kB, mediated by IL-8, promotes the switch to a pro-tumor neutrophil profile [15]. Conversely, inhibitors of CXCR2 have been shown to reduce tumor-associated neutrophil functions such as NETs in melanoma [16]. Moreover, in tumor models, the inhibition of CXCR2 has been linked to an increased response to immunotherapy, suggesting that the latter may potentially benefit from a combination with CXCR2 inhibitors [17].

The apparently linear model of ontogenesis described above does not reflect the actual complexity of neutrophil states and subsets that have been described in the literature. In cancer biology, neutrophil heterogeneity underlies the differential roles of these cells and impacts the feasibility of therapeutic strategies targeting all or only individual neutrophil subsets. Neutrophil heterogeneity and function have been related to general host factors: infection, including severe COVID-19 [18], chronic inflammation, obesity, smoking and other stressors [19]. As far as locoregional factors are concerned, it is currently unknown whether there is a tissue-specific behavior of neutrophils that may differentially influence cancer arising in different organs.

Neutrophils have also been credited with protective functions related to an anti-tumorigenic phenotype [20]. This apparent paradox may be justified by specific neutrophil states associated with tumor stages and early or advanced disease [21,22]. In melanoma, neutrophil progenitors are elevated in patients’ blood samples as well as in the bone marrow, blood and tumor of melanoma models, where they partner with high PD-L1 expression to promote neoplastic growth [23]. These observations suggest that neutrophils may be mobilized from the bone marrow at different developmental and functional stages, and support apparently opposite roles in tumor growth due to neutrophil heterogeneity.

Moreover, cells of the innate immune system can become trained after host exposure to agonists with anti-tumor effects, such as beta-glucans [24], adding additional and to date unknown levels of complexity to this system. Recent evidence has drawn attention to the role of melanoma stem cells, a subpopulation of self-renewing tumor cells, that establish a two-way crosstalk with neutrophils, whereby cancer stem cells are able to educate neutrophils to support melanoma progression and immune escape [25], and tumor-infiltrating neutrophils induce stemness in melanoma cells, secrete pro-inflammatory cytokines and suppress cytotoxic T cells [26].

Finally, though our current knowledge is informative of associations between neutrophil states and tumors, mechanistic evidence on their causative role in tumor promotion is still incomplete. Moreover, an in-depth understanding of neutrophil heterogeneity in cancer will be required to devise targeted strategies aimed at specific neutrophil subtypes relevant to cancer.

## 3. Neutrophils as Biomarkers in Melanoma Patients Treated with Immune Checkpoint Inhibitors

A substantial fraction of subjects with advanced melanoma is not responsive to immune checkpoint inhibitors, and, in the era of precision medicine, the identification of predictive biomarkers is a current unmet need. Ideal biomarkers are expected to specifically inform on the response and tolerability to treatment while being easily measurable in the clinical setting. Candidate tumor markers related to the mechanism of action of PD-1 inhibitors, such as PD-L1 tissue expression, immune cell infiltrate and tumor mutation burden are not reliable predictors of a response to immunotherapy in melanoma [27,28]. Consequently, recent research has shifted from these tissue markers to host factors that carry the additional advantage of being easily measured in the serum.

Serum lactate dehydrogenase is an established prognostic factor in metastatic melanoma and its role as a predictive marker has been recently investigated in a meta-analysis [29].

Elevation in peripheral blood neutrophils, measured as an absolute count and as a neutrophil-to-lymphocyte ratio (NLR), has been assessed as a potential biomarker in melanoma. A 2018 study by Capone et al. showed significantly lower progression-free survival (PFS) and overall survival (OS) with an NLR ≥ 5 than with a low NLR (2.0 vs. 9.0 and 2.9 vs. 16.0 months, respectively) in melanoma patients receiving nivolumab. An improved response to treatment was also observed in the low NLR group [30]. Another study conducted by Garnier et al. in subjects with melanoma treated with anti-PD-1 immunotherapy showed that the NLR was a prognostic biomarker independent of other known prognostic factors, such as LDH [31]. In a similar study by Bartlett et al., the authors performed a dynamic measurement of the NLR highlighting an association between an NLR increase of at least 30% after two cycles of immunotherapy and a lack of response to treatment [32]. A study by Qi et al. showed comparable results in Chinese subjects with melanoma treated with PD-1 inhibitors. Using an NLR cut-off of 3, the authors demonstrated higher PFS and OS in subjects with a low NLR compared to those with a high NLR (7.0 vs. 2.2 and 18.0 vs. 5.6 months, respectively) [33]. A 2022 study by Koczka et al. showed statistically shorter PFS and OS in subjects with neutrophilia compared to those with neutrophils below the upper limit of normal (2.1 vs. 7.1 and 2.9 vs. 15.6 months, respectively). Moreover, an elevated NLR ≥ 4 was associated with inferior PFS and OS compared to a ratio <4 (2.5 vs. 8.3 months and 5.2 vs. 19.4 months, respectively). In this case, the authors reported similar response rates to PD-1 inhibitors irrespective of neutrophilia and the neutrophil-to-lymphocyte ratio [34]. These findings were included in a recent meta-analysis that additionally highlighted that the prognostic value of NLR may change in different geographic regions. This may be due to the differences in both tumor characteristics and immune functions between regions. On the one hand, certain subtypes with specific genetic signatures, such as acral and mucosal melanoma associated with KIT mutations, are most common in Asian subjects [35]; on the other hand, there may be a heterogeneity in lymphocyte number with reportedly lower counts in Asian populations [36]. Finally, a recent study, conducted on a large population of 272 subjects with BRAF wild-type metastatic melanoma treated with first-line immune checkpoint inhibitors, demonstrated that the baseline values of blood cells as well as their early variation after one month of treatment are predictors for the efficacy of immunotherapy. Specifically, the authors established a negative correlation between PFS and OS for higher neutrophils and the NLR at baseline as well as for higher NLR variation [37].

These studies suggest that markers of general immune function and systemic inflammatory response such as the NLR are related to survival in subjects treated with PD-1 inhibitors. Moreover, measurement of the NLR in the serum is straightforward and economic, enabling repeatable dynamic sampling. Further investigation is required to establish their predictive role to aid in the patient selection of subjects that would maximally respond to immunotherapy and, conversely, to spare those who would not benefit from treatment from the risk of toxicity. Finally, the immune mechanism underlying the relation between a high NLR and worse clinical outcome in melanoma is still unknown.

## 4. Neutrophils as Therapeutic Targets in Dermatology

Granulocyte and monocyte apheresis (GMA) is an extracorporeal adsorptive apheresis technique which has been proposed in dermatology as an innovative alternative treatment option for disorders mediated by active neutrophils [38]. A growing interest for this procedure is motivated by its non-pharmacological nature that makes it particularly useful and tolerable in subjects with chronic conditions. The usual schedule of GMA consists of weekly 60-min sessions that can be repeated five to ten times, according to the severity of the disease and to the clinical response [39]. A total of 1800 mL of blood per session are drawn from a cubital vein at a 30 mL/min flow rate, transit through a column containing absorptive cellulose acetate beads and are then reintroduced in the contralateral cubital vein [40]. Anticoagulation, usually with nafamostat mesylate or heparin, is necessary throughout the procedure [41].

The effects of GMA are due to a selective depletion of activated cells of the myeloid lineage [42]. The molecular mechanisms of action of this procedure are to date only partially understood. Cellulose acetate beads are able to activate and absorb the iC3b complement component. This is considered the main mechanism for selectivity since iC3b is the ligand for the integrin family molecule Mac-1 which is expressed on the surface of activated granulocytes and monocytes/macrophages [39]. Activated neutrophils are trapped in the column by binding Mac-1 and Fc-gamma receptors, which interact with IgG on the beads. The removal of Mac-1, expressing activated neutrophils, reduces tissue infiltration by these cells to sites of inflammation and additionally increases the percentage of immature granulocytes that are physiologically scarcely inflammatory [43,44]. Overall, the procedure achieves a 65% reduction of activated granulocytes from the circulation [42].

The principal advantage supporting the use of GMA is its excellent safety. Adverse events reported in the literature were related to obtaining a venous access and maintaining an adequate flow, risk of coagulation, increased venous pressure and difficulty in venous return, and none was considered serious [42,43]. The procedure is overall well tolerated by patients with reports of mild symptoms related to the procedure including weakness, headache, chills and fever [45]. Finally, GMA has not been related to an increased risk of infection since the related depletion of neutrophils does not cause a state of immunodeficiency [42]. This favorable profile has encouraged the use of GMA together with systemic treatments, with the aim of achieving a faster response and higher efficacy [46]. The cost of GMA was estimated to be almost EUR 7000 for five sessions; future studies with a long follow-up are required to establish the optimal therapeutic regimens and the long-term efficacy of GMA, and to ultimately assess its cost-effectiveness in different settings [47].

Past studies provided preliminary evidence on the effect of GMA in the setting of advanced cancer associated with increased granulocytes. In a 1995 trial, seventeen subjects with recurrent metastatic tumors underwent fifteen sessions of GMA, showing a partial response in four cases and no significant side effects [48]. These results led the authors to suggest that GMA may improve the quality of life in these patients and be particularly useful as a part of combined oncologic treatments [48,49]. A 1996 pilot study conducted on two patients with recurrent metastatic tumors showed that a course of fifteen GMA sessions reduced tumor size and prolonged patient survival with no serious adverse events [50].

## 5. Conclusions and Future Directions

In melanoma, an increased presence of cancer-associated neutrophils has been related to an unfavorable prognosis and poor treatment outcomes [51].

This informative association constitutes the basis for expanding our knowledge on the role of neutrophils in cancer immunology and for discovering neutrophil-targeted immunotherapies [5]. Specific neutrophil subtypes within the TME play a key role in shifting the cancer-immune set point from an anti-tumor to a pro-tumor phenotype [27]; novel approaches targeting cancer-associated neutrophils may be devised to reshape TME-promoting tumor recognition by T lymphocytes and natural killer cells [4]. In the context of advanced melanoma, we suggest that future investigations assess the use of GMA in combination with cancer immunotherapy. Moreover, we suggest that the addition of this non-pharmacological neutrophil-depleting technique be investigated in subjects that do not respond adequately to therapy with immune checkpoint inhibitors; in such patients, GMA may theoretically mount the host immune response to surpass a threshold for boosting the effect of immunotherapy.

## Data Availability

Not applicable.

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
