# Peer review of "Granulocyte Apheresis: Can It Be Associated with Anti PD-1 Therapy for Melanoma?"

_medicina, 2022, doi:10.3390/medicina58101398_

Round 1
Reviewer 1 Report
Authors reported basic understanding on the role of neutrophils in cancer biology, prognosis and response to immunotherapy in cancer, particularly in advanced melanoma.
They also propose that granulocyte monocyte apheresis could become an investigative treatment targeting melanoma-associated neutrophils potentially to be used in combination with immune checkpoint inhibitors.
The paper is a concise but comprehensive overview of the role of neutrophils in cancer progression and an open window to new perspectives on cancer treatment by intervening in the manipulation of these cells.
Main concerns: the Authors suggest investigations with the use of this non-pharmacologic neutrophil-depleting technique in combination with cancer immunotherapy.
this is currently only a recondite hope of the authors, but on what preclinical basis does this hypothesis rest?
This aspect is in my opinion to be investigated further.
Minore concerns:
Session 3. Neutrophils as biomarkers in melanoma patients treated with immune checkpoint inhibitors.
1. Line 114. Correct "serum neutrophils" to "peripheral blood neutrophils" or something similar.
2. Please, add our recent data in regarding a large Italian population of 272 BRAF wild-type metastatic patients treated with first line checkpoint inhibitors.
Reference: Guida M, Bartolomeo N, Quaresmini D, Quaglino P, Madonna G, Pigozzo J, Di Giacomo AM, Minisini AM, Tucci M, Spagnolo F, Occelli M, Ridolfi L, Queirolo P, De Risi I, Valente M, Sciacovelli AM, Chiarion Sileni V, Ascierto PA, Stigliano L, Strippoli S. Basal and one-month differed neutrophil, lymphocyte and platelet values and their ratios strongly predict the efficacy of checkpoint inhibitors immunotherapy in patients with advanced BRAF wild-type melanoma. J Transl Med. 2022 Apr 5;20(1):159. doi: 10.1186/s12967-022-03359-x. PMID: 35382857; PMCID: PMC8981693.
Author Response
Thank you for your efforts on our paper and for your constructive comments. We have carefully considered each issue raised and revised our manuscript accordingly:
- Thank you for your constructive comment, we have reported the preliminary results of past studies available in the literature. Please see page 4 lines 253-260 and references [48-50]: “Past studies provided preliminary evidence on the effect of GMA in the setting of advanced cancer associated with increased granulocytes. In a 1995 trial, 17 subjects with recurrent metastatic tumors underwent 15 sessions of GMA showing partial response in 4 cases and no significant side effects [48]. These results led the authors to suggest that GMA may improve quality of life in these patients and be particularly useful as a part of combined oncologic treatments [48,49]. A 1996 pilot study conducted on two patients with recurrent metastatic tumors showed that a course of 15 GMA sessions reduced tumor size and prolonged patient survival with no serious adverse events [50].
- Thank you, we have corrected wording according to your suggestion; please see page 3 line 165: “Elevation in peripheral blood neutrophils…”
- Thank you for your suggestion: we have added the results of this recent study and the relevant reference. Please see page 3 lines 191-196 and reference [37]: “Finally, a recent study conducted on a large population of 272 subjects with BRAF wild-type metastatic melanoma treated with first line immune checkpoint inhibitors demonstrated that baseline values of blood cells as well as their early variation after one month of treatment are predictors for the efficacy of immunotherapy. Specifically, the authors established a negative correlation with PFS and OS for higher neutrophils and NLR at baseline as well as for higher NLR variation [37].”
Guida, M.; Bartolomeo, N.; Quaresmini, D.; Quaglino, P.; Madonna, G.; Pigozzo, J.; Di Giacomo, A. M.; Minisini, A. M.; Tucci, M.; Spagnolo, F.; et al. Basal and one-month differed neutrophil, lymphocyte and platelet values and their ratios strongly predict the efficacy of checkpoint inhibitors immunotherapy in patients with advanced BRAF wild-type melanoma. J Transl Med 2022, 20, 159. https://doi.org/10.1186/s12967-022-03359-x.
Reviewer 2 Report
The manuscript deals with an interesting topic with possible clinical implications. Howwever, revisions are needed to make it worthy of publication:
-The introduction is clear and well structured, however some additional insights on the role of PD1, could improve it and make it reacher and more complete. If they consider it appropriate, the authors could add PMID: 33261292.
-The remainder of the manuscript is very thorough and accurate and the conclusions are in agreement with it.
Author Response
Thank you for your careful review of our paper and for your encouraging comments. Please find our responses below:
- Thank you for your suggestion. We have added a relevant statement to the introduction. Please see page 1 lines 25-30 and reference [3]: “Melanoma is rightfully considered an immune-responsive malignancy: the development and progression of this cutaneous tumor is strongly driven by down-regulators of the immune system, such as PD-1 and its ligand, that constitute cancer mechanisms for immune evasion. Inhibitors of the PD-1 immune checkpoint proved able to overcome immune escape and reestablish the antitumor activity of melanoma-infiltrating T lymphocytes [3].”
- Perisano, C.; Vitiello, R.; Sgambato, A.; Greco, T.; Cianni, L.; Ragonesi, G.; Malara, T.; Maccauro, G.; Martini, M. Evaluation of PD1 and PD-L1 expression in high-grade sarcomas of the limbs in the adults: possible implications of immunotherapy. J Biol Regul Homeost Agents 2020, 34, 289–294.
Round 2
Reviewer 1 Report
Thanks a lot for your excellent work